# NEOCORTICAL CELL TYPE CLASSIFICATION FROM ELECTROPHYSIOLOGY RECORDINGS USING DEEP NEURAL NETWORKS

## ABSTRACT

Understanding the neural code requires identifying different functional units involved in the neural circuits. One way to identify these functional units is to solve a neuron type classification problem. For decades, current-clamp electrophysiology recordings have provided the means to classify the neurons based on subtle differences in action potential shapes and spiking patterns. However, significant variations in neuronal type definitions, classification pipelines, and intrinsic variability in the neuronal activities make unambiguous determination of neuron type challenging. Previous solutions to this electrophysiology-based cell type classification problem consisted of dimensionality reduction juxtaposed with clustering using hand-crafted action potential features. Recent discoveries have allowed genetics-based cell-type classifications, which have fewer ambiguities, but they are less practical in vivo and have even lower throughput. Leveraging the unprecedented ground truth data published in the Allen Institute Cell Types Database, which contains anatomical, genetic, and electrophysiological characterizations of neurons in the mouse neocortex, we construct a robust and efficient convolutional neural network (CNN) that successfully classifies neurons according to their genetic label or broad type (excitatory or inhibitory) solely using current-clamp electrophysiology recordings. The CNN is configured as a multiple-input single-output network consisting of three subnetworks that take in the raw time series electrophysiology recording as well as the real and imaginary components of its Fourier coefficients. Our single pipeline method is fast and streamlined while simultaneously outperforming a previous method. Furthermore, our method achieves classification with more classes using only a single current-clamp time series trace as the input. This end-to-end convolutional neural network-based classification method removes the need for hand-crafted features, specific knowledge, or human intervention for quick identification of the neocortical cell type with high accuracy, enabling interpretation of experimental data in a bias-free manner and understanding of a much broader scientific context.

## 1 INTRODUCTION

The neuronal type classification problem has been present in neuroscience since Ramón y Cajal's presentation of the neuron doctrine, which highlighted the ample diversity of neurons. Neuroscientists hypothesized that morphological differences play a functional role in neural circuits. This intuition was extended to the investigation of the differences in neuronal activity with the appearance of the current clamp technique, which allowed observations of various action potential shapes and patterns in neurons.

Features such as action potential (AP) threshold and frequency, full width at half maximum (FWHM) of an action potential (AP width), or afterhyperpolarization values and their common ratios in subsequent action potentials in a train, which are easily distinguishable by a person, aimed to describe the differences among the observed variabilities in neuronal activities. Although no two neurons have the same activity, these hand-crafted features were used to define the electrophysiological types of neurons (Beierlein et al., 2003; Nowak et al., 2008). Due to the intrinsic neuron-to-neuron variabil-

ities and the lack of established features that definitively separate neuron types, no single pipeline is sufficient enough for unambiguous classification.

Several electrophysiology-based classification pipelines including methods that consider more features and ones that focus only on a single neuronal subpopulation coexist today (Petilla Interneuron Nomenclature Group; PING and others, 2008; Markram et al., 2004). Previous solutions to this cell type classification problem were comprised of a combination of dimensionality reduction and clustering using calculated action potential features and cell morphology. These approaches suffered in classification accuracy as they relied on AP shape, spiking pattern, or cell shape parameters that span a continuous feature space, which often do not have clear separation boundaries.

In addition to the aforementioned classification method based on morphology and electrophysiology, one based on the genetic makeup of neurons appeared recently in systems neuroscience (Tasic et al., 2018). A vast majority of neurons can be clustered according to their genetic encoding characteristic of the group proteins. This gene-based consideration of neuron type allows a less ambiguous classification pipeline. The most commonly used genetic types of neurons are neuron-derived neurotrophic factor (Ndnf), parvalbumin (Pvalb), somatostatin (Sst), and vasoactive intestinal peptide (Vip) interneurons, and excitatory (Exc) neurons that are predominantly pyramidal cells.

Furthermore, the emergence of genetically modified transgenic lines of animals, where cells of a selected genetic type can be marked by a fluorophore, enabled studies relating electrophysiological neuronal activity to the genetic type of neurons (Taniguchi et al., 2011). Attempts to characterize the electrophysiological features of specific genetic types of neurons revealed that the majority of Pvalb interneurons correspond to Fast Spiking (FS) electrophysiological cell type, but both Vip and Sst interneurons as well as excitatory neurons can appear as Regular Spiking (RS) or Low Threshold Spiking (LTS) types (Contreras, 2004). Therefore, we can conclude that there does not exist a clear mapping from one classification scheme to another, partially due to existing methods' low throughput and differences in definitions and experimental pipelines.

To address the challenges presented by the above neuron type classification problem and to improve on the existing classification pipeline architectures, a robust and efficient convolutional neural network (CNN) that successfully classifies neurons according to their genetic label or broad type (excitatory or inhibitory) solely using short snippets of current-clamp electrophysiology recordings has been developed. The method presented in this paper relies on the ground truth data published in the Allen Institute Cell Types Database, which contains anatomical and morphological descriptions, genetic types, and electrophysiology features of thousands of neurons in the mouse neocortex (Gouwens et al., 2019). We use this open access database, which is one of the flagship initiatives of the Allen Institute for Brain Science, to obtain the genetic type label of a given neuron, bypassing supervised definitions of all signal features. The following sections detail the related state-of-the-art methods, the specifications of our novel deep neural network-based neuron type classification method, as well as the results obtained using our custom CNN architecture.

## 2 RELATED WORK

Gouwens et al. (2019) built and made publicly available the Allen Institute Cell Types Database. From this database, 17 electrophysiological, 38 morphological, and 46 morpho-electric neuron types were identified using a custom classification pipeline. The authors employed biocytin-based neuronal reconstruction to extract morphological features and used raw current-clamp electrophysiology recordings of cells from the mouse visual cortex in vitro as inputs for electrophysiological features. After computing handcrafted single action potential features including action potential height, threshold, upstroke speed, and downstroke speed, as well as features corresponding to action potential trains, such as interspike intervals and spiking frequency, principal component analysis and t-distributed stochastic neighbor embedding techniques were applied to project the high-dimensional electrophysiological feature space into two dimensions. With clustering, the authors were able to identify 17 electrophysiological neuron types, 4 of which were classified as excitatory subtypes and 13 of which were inhibitory. The 13 inhibitory subtypes were further mapped to the four inhibitory interneuron types based on their genetic tags: Sst, Vip, Pvalb, and Ndnf.

Ghaderi et al. (2018) successfully developed a semi-supervised technique to classify neuron types using limited in vivo electrophysiology recordings data. The authors considered only 3 types of neurons: excitatory pyramidal (Pyr) cells, parvalbumin positive (Pvalb) interneurons, and somatostatin positive (Sst) interneurons from layer 2/3 of the mouse primary visual cortex. After extracting single spikes, they extracted discriminative action potential features by computing the Discrete Cosine Transform of the recorded electrophysiology traces. Principal component analysis and fuzzy c-means clustering were then performed, and neurons were subsequently classified using a minimum distance classifier. The authors achieved accuracies of $91.59 \pm 1.69$, $97.47 \pm 0.67$, and $89.06 \pm 1.99$ for Pvalb, Pyr, and Sst, respectively, which yielded an overall accuracy of $92.67 \pm 0.54\%$. This classification algorithm pipeline was further applied to the in vitro data from the Allen Institute Cell Types Database containing Pvalb, Sst, 5HT3a, and Vip genetic types. Testing on a dataset comprised of a pool of 50 neurons where multiple electrophysiology traces have been recorded for each neuron, the authors achieved accuracies of $93.57 \pm 0.59\%$, $89.15 \pm 0.63\%$, $81.69 \pm 0.56\%$, $79.23 \pm 1.38\%$, and $77.02 \pm 0.91\%$ for Pvalb, Sst, Vip, 5HT3a, and Pyr, respectively.

## 3 METHODS

In this paper, we present a neuron type classification technique based on a simple convolutional neural network (CNN) architecture. Using in vitro current-clamp electrophysiology recording traces of neurons in the mouse neocortex, the CNN is configured as a multiple-input single-output network consisting of three subnetworks. The first subnetwork takes in a portion of the raw time series recording that is 50 ms in duration and contains at least one action potential. The real and imaginary components of a trace's Fourier coefficients are fed into the second and third subnetworks of the CNN, respectively. A graphical representation of the CNN architecture is shown in Figure 3.

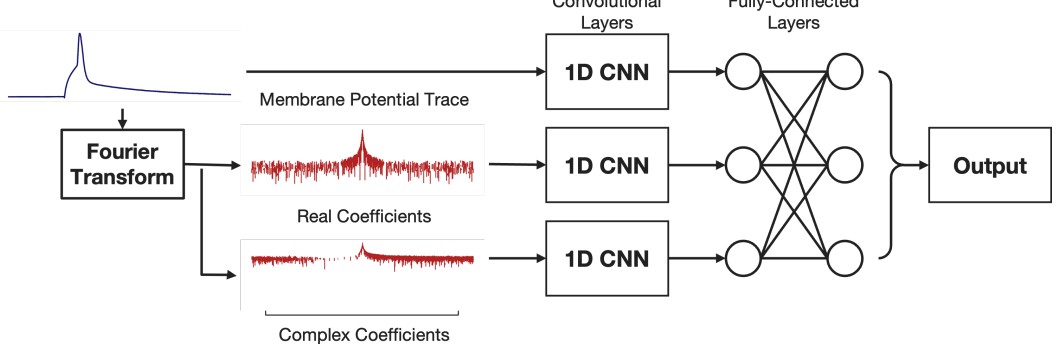

Figure 1: Graphical representation of the triple-input CNN architecture.

### 3.1 DATASET

We use data collected from 1947 cells in the Allen Institute Cell Types Database (Gouwens et al., 2019). Of these, we omitted 81 cells containing only morphological features and lacking electrophysiology recordings, and only the remaining 1866 cells containing electrophysiology recordings were used to build our training, validation, and test sets. Every one of these 1866 cells was obtained from transgenic animal lines, and thus is associated with a genetic label (Exc, Ndnf, Vip, Sst, and Pvalb). Each neuron contains approximately 50 electrophysiology trace recordings that are responses to multiple current-clamp stimuli including short square, long square, ramp, and noise. Of these traces, we only considered responses to the short square stimulus, which are 3 ms in duration, which is just long enough to induce a single action potential. Each of these recordings is collected at either 200 kHz or 50 kHz sampling rate. Because each neuron is recorded at a different stimulus level, we only consider the traces that contain an action potential.

To obtain the most useful information about the neuron type being assessed, the type of classification task that our architecture solves can be dictated by the needs of a neurophysiologist. One task is to distinguish neuronal activity coming from an excitatory (Exc) or an inhibitory (Inh) neuron.

This task is less informative, but provides explanation useful for analyzing electrophysiological recordings. The more interesting task is to discriminate among the 5 genetic types (Pvalb, Sst, Vip, and Ndnf inhibitory interneuron types and Exc neuron type), which constitute the broader excitatory and inhibitory categories. Reliable and accurate classification among the 5 genetic types is highly valuable to the neuroscience community. We train our network to perform the two aforementioned tasks.

### 3.2 PREPROCESSING

One of the main objectives of our approach is to remove as much supervised overhead in the data processing stage of the neuron type classification pipeline as possible. The only preprocessing steps we perform on the raw time series data is removing excess temporal portions of recordings that do not provide useful information. We are only interested in the portion of the recording containing an action potential, so only 50 ms of recording, 25 ms before and 25 ms after the onset of the short square stimulus, was considered. The 25 ms of pre- and post-stimulus time duration was chosen to ensure that potential discriminative features that may be present before or after the onset of the stimulus would be captured by the representation learning performed by our convolutional neural network. The 25 ms of post-stimulus time guarantees that a single action potential has returned to its resting membrane potential after depolarization and hyperpolarization.

We also take the fast Fourier transform of these 50 ms time series traces. The real and imaginary components of the resulting Fourier coefficients, in addition to the corresponding time series traces, are used as inputs to the subnetworks of our convolutional neural network architecture.

For model selection and performance, we divide the aforementioned dataset collection from the Allen Institute Cell Types Database based on the unique cell identification numbers. The ratio between the training and validation sets was fixed at 8:2. Once the best generalized performing model was identified, we independently split the dataset again based on the unique cell identification numbers. Eighty percent of the data were reserved for training and the remaining 20% were set aside as the test set data. The test set was further split as follows: 80% test and 20% validation. The validation set was used to tune the network hyperparameters. This dataset separation by cell prevents overfitting, and provides both significant advantage and improvement to the existing methods. Due to natural cell-to-cell variations, basic action potential features like resting membrane potential can vary considerably among cells of the same genetic type. We therefore use and report the maximum validation accuracy we obtain over 100 epochs.

### 3.3 NETWORK ARCHITECTURE

We use a multiple-input single-output convolutional neural network (CNN) for training. To remove the need for handcrafted features, our deep neural network uses a one-dimensional convolutional neural network as a feature encoder and employs dense layers to output class predictions. The standard one-dimensional CNN encoder is implemented using PyTorch. The encoder contains 6 convolutional layers, and each layer is passed through a Rectified Linear Unit (ReLU) activation function. Batch normalization introduced by Ioffe & Szegedy (2015) is applied to each activated layer. The exact specifications of each layer is shown in c) of Figure 3.3.

For training, we employ the Adam optimizer—with the learning rate set to $10^{-3}$ and the $\ell_2$ regularization parameter set to $10^{-5}$—to minimize the cross entropy loss with sum reduction (Kingma & Ba, 2014). The initial weights were randomly generated.

An 8:2 ratio training-validation data set split was used to select the optimal network model configuration. We first tried a single-input single-output configuration, where only the top stream in c) of Figure 3.3 was used. This architecture resulted in a validation accuracy of 88.52% for classification among the 5 genetic neuron types: Exc, Pvalb, Sst, Ndnf, and Vip; and a validation accuracy of 96.35% for classification between excitatory and inhibitory broad types.

We then tested using a dual-input single-output configuration. Features were trained independently on each subnetwork and were concatenated at the final step for classification. Using only the real component of the Fourier coefficients as input to the additional subnetwork resulted in validation accuracies of 91.43% for classification among the 5 genetic neuron types, and 99.38% for classification between excitatory and inhibitory broad types. Similarly, using only the complex component

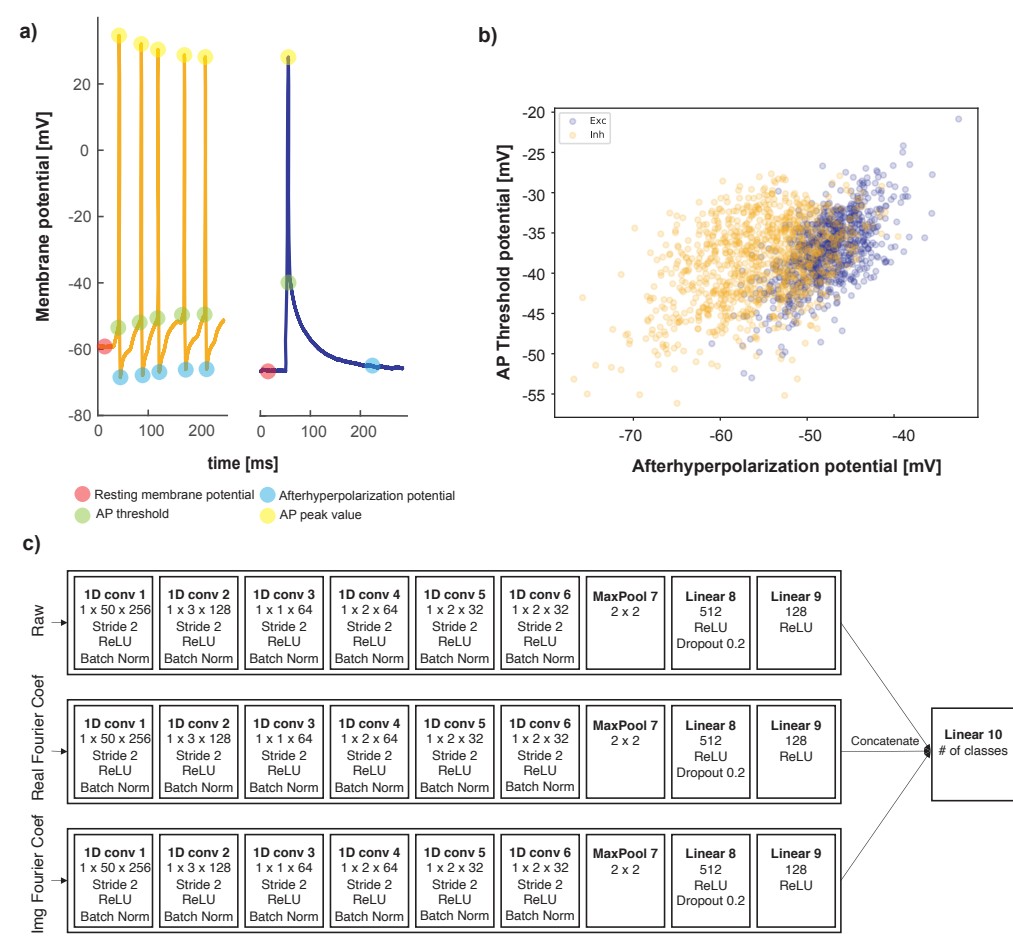

Figure 2: **a)** Canonical examples of neuronal membrane potential traces featuring typical FS, Inh neuron (left) and RS, Exc neuron (right). Some exemplary AP features are marked on the traces. **b)** Example plot showing continuous values of selected AP features characteristic of Exc and Inh groups of neurons. **c)** Detailed schematic diagram of the CNN architecture design developed and described in this paper.

of the Fourier coefficients resulted in validation accuracies of 89.13% for classification among the 5 genetic types, and 96.58% for classification between excitatory and inhibitory broad types.

Finally, we used a triple-input single-output configuration where the raw current-clamp electrophysiology recording time series trace as well as both the real and imaginary components of the Fourier coefficients are used as inputs. Features were also trained independently on each subnetwork and were concatenated at the final step for classification. This configuration was our best performing architecture, which resulted in validation accuracies of 92.05% for classification among the 5 genetic types and 98.10% for classification between excitatory and inhibitory broad types. This triple-input network architecture's $\ell_2$ regularization hyperparameter was tuned to $10^{-5}$.

### 3.4 TRANSFER LEARNING

One of the biggest challenges in decoding the neural circuit and in systems neuroscience in general is obtaining sufficient amount of data for each neuron type. Supervised learning requires a large number of sample data points to train efficient classifiers. One of the main difficulties we encountered in this work was the problem of class imbalance present in the datasets. For example, there are many more data points from excitatory (466) and parvalbumin (241) cells than Ndnf (46), Sst (86), and Vip (45) cells in the 5 class validation set; and many more data points from excitatory (644)

and parvalbumin (260) cells than Ndnf (38), Sst (130), and Vip (72) cells in the 5 class test set. We attempt to solve this problem using transfer learning.

From the results of our triple-input single-output network, we can conclude that the 2 class classification task of identifying excitatory versus inhibitory cells achieves high accuracy. This observation inspired us to first train the model on the task it does well, and then conduct fine-tuning. We therefore trained the aforementioned CNN network for the 2 broad type (Exc and Inh) classification task first for 100 epochs. We then changed the task to the 5 genetic type (Exc, Pvalb, Sst, Ndnf, and Vip) classification using the network trained on the two broad type classification task. The network uses the same CNN weights from the 2 type classification task, but has an output layer changed to the 5 type classification task. Then, the updated network underwent fine-tuning on the 5 type classification task for 100 epochs.

The same Adam optimizer with the learning rate of $10^{-3}$ and $\ell_2$ regularization parameter of $10^{-5}$ was used for training. In order to investigate the model's ability to undergo fine-tuning on a limited number of data points, we subjected the network to endure fine-tuning on the following training-test data set split ratios: 3:7, 5:5, and 8:2. For the 5 class classification task, at the epoch corresponding to the highest validation accuracy, the computed test set precisions were 86.87%, 87.46%, and 89.22%, for the above three training-test set split ratios, respectively. Table 1 reports the details of the test set precision values of the above transfer learning.

Table 1: Transfer learning precision

| Type\Training-Test Set Ratios | 3:7 | 5:5 | 8:2 |
| --- | --- | --- | --- |
| Exc | 97.65 | 98.08 | 94.60 |
| Ndnf | 96.59 | 89.43 | 90.00 |
| Pvalb | 80.99 | 79.93 | 91.25 |
| Sst | 71.76 | 72.26 | 84.13 |
| Vip | 44.65 | 51.70 | 45.65 |
| **Weighted Avg** | 86.87 | 87.46 | 89.22 |

In addition, we fine-tuned the network on the 5 type classification task, used a 5 type classification validation set, selected the epoch with the highest validation accuracy, and evaluated a 2 class classification task's test set accuracy. This particular network, having never observed this 2 class classification test set data during training, resulted in a test set accuracy of 98.30%.

Finally, we trained the network on the current-clamp electrophysiology recordings data sampled at 200 kHz. We then transferred the trained model to a task using 50 kHz recordings while keeping the input size constant. This resulted in action potential features being stretched or compressed to fit the new sampling rate. When comparing this transfer approach to a control experiment where a newly initialized network was trained from start on the 50 kHz sampling rate data, we observed that the transferred model required less data to achieve higher accuracy values, as shown by Figure 3.4.

This finding highlights an important advantage of our model. Collecting data at different sampling rates is expensive. We have demonstrated that our method is capable of rapidly adapting to data obtained from varying sampling rates. We can thus conclude that our method significantly reduces costs associated with data collection.

## 4 RESULTS

Our best performing triple-input single-output network architecture resulted in a test set weighted average precision of 89.76% for classification among the 5 genetic neuron types: excitatory (Exc), parvalbumin (Pvalb), somatostatin (Sst), neuron-derived neurotrophic factor (Ndnf), and vasoactive intestinal peptide (Vip) cells. This finalized network architecture resulted in a test set weighted average precision of 98.28% for classification between excitatory and inhibitory broad types. For both tasks, the individual classes' precisions, recalls, and $F_1$-scores are reported in Table 2.

In order to obtain a better understanding and aid interpretablity of the triple-input CNN's classification performance, we investigated which genetic type was outputted for every genetic type input in

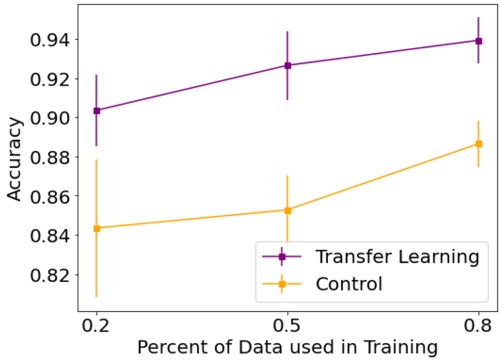 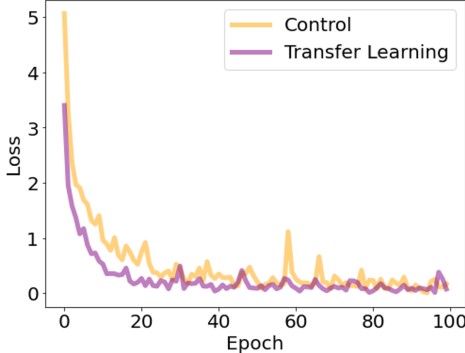

Figure 3: Performance comparison between models trained from start using data sampled at 50 kHz and models transferred from a 200 kHz sampling rate data task to a 50 kHz sampling rate data task. Transferred models achieve higher accuracy with less data, suggesting efficient learning and adaptation of action potential features across multiple sampling rates.

Table 2: Triple-input CNN precision

| Type | Precision (%) | Recall (%) | $F_1$-Score (%) | Support |
|---|---|---|---|---|
| **2 Class Validation Set** | | | | |
| Exc | 98.78 | 98.78 | 98.78 | 490 |
| Inh | 98.52 | 98.52 | 98.52 | 406 |
| **Weighted Avg** | 98.66 | 98.66 | 98.66 | 896 |
| | | | | |
| **2 Class Test Set** | | | | |
| Exc | 97.28 | 99.51 | 98.38 | 611 |
| Inh | 99.42 | 96.80 | 98.10 | 532 |
| **Weighted Avg** | 98.28 | 98.25 | 98.25 | 1143 |
| | | | | |
| **5 Class Validation Set** | | | | |
| Exc | 96.27 | 99.79 | 98.00 | 466 |
| Ndnf | 84.62 | 95.65 | 89.80 | 46 |
| Pvalb | 99.17 | 98.76 | 98.96 | 241 |
| Sst | 88.24 | 87.21 | 87.72 | 86 |
| Vip | 87.50 | 46.67 | 60.87 | 45 |
| **Weighted Avg** | 95.23 | 95.36 | 94.94 | 884 |
| | | | | |
| **5 Class Test Set** | | | | |
| Exc | 94.93 | 98.76 | 96.80 | 644 |
| Ndnf | 57.63 | 89.47 | 70.10 | 38 |
| Pvalb | 93.08 | 93.08 | 93.08 | 260 |
| Sst | 77.54 | 82.31 | 79.85 | 130 |
| Vip | 70.59 | 16.67 | 26.97 | 72 |
| **Weighted Avg** | 89.76 | 90.12 | 88.75 | 1144 |

the 5 type classification task. The results are summarized in Table 3 and are represented graphically by the confusion matrix in Figure 4.

Although not necessarily a direct comparison of performance, Ghaderi et al. (2018) achieved an overall accuracy of 84.13% using the Allen Institute Cell Types Database containing Pyr (Exc), Pv (Pvalb), Sst, 5ht3a, and Vip cells. Using the Allen Institute Cell Types Database containing Pyr (Exc), Pv (Pvalb), Sst, Ndnf, and Vip cells, we achieved an overall accuracy of 90.12%.

Table 3: Classification accuracy results (%)

| Output\Input | Sst | Vip | Ndnf | Pvalb | Exc |
|---|---|---|---|---|---|
| Sst | **70.62** | 29.39 | 0.25 | 5.8 | 0.49 |
| Vip | 13.0 | **34.92** | 8.65 | 2.52 | 1.29 |
| Ndnf | 0.18 | 5.63 | **85.24** | 0.83 | 0.14 |
| Pvalb | 10.1 | 10.38 | 4.45 | **90.82** | 0.01 |
| Exc | 6.1 | 19.68 | 1.42 | 0.03 | **98.08** |

Table 4: For each cell type, both the correct classification rate as well as the rates for all misclassified genetic types are shown. Each column represents an input cell type, and each row indicates how frequently the CNN misclassified the input cell type as one of the other cell types.

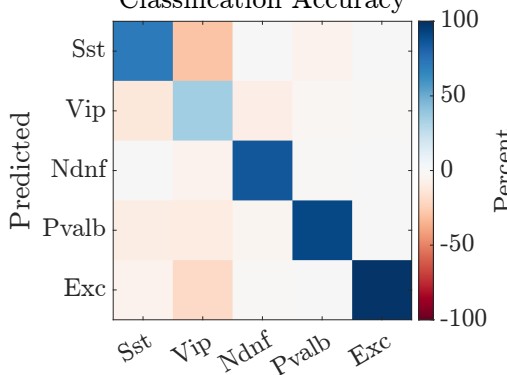

Figure 4: Confusion matrix of classification accuracy scores. Misclassification rates are represented by negative percentage values while correct classification rates are represented by positive percentage values.

For a more direct comparison of performance, we trained and applied the triple-input CNN on a partial dataset containing only Pv (Pvalb), Pyr (Exc), and Sst cells, and achieved the accuracies reported in Table 5. Our triple-input CNN was able to outperform the existing method on all fronts.

Table 5: Overall accuracy results comparison (%)

| Class | Ghaderi et al. (2018) | Triple CNN |
|---|---|---|
| Pv | 91.59 | **95.95** |
| Pyr | 97.47 | **98.29** |
| Sst | 89.06 | **90.00** |
| **Overall** | 92.67 | **96.89** |

## 5 CONCLUSION AND FUTURE WORK

We presented in this paper a solution to the neuron classification problem that avoids the use of existing imperfect, non-standardized, and cumbersome electrophysiological classification schemes. The triple-input CNN architecture described in this paper maps the pool of neurons into less ambiguous genetic classifications that is normally not widely accessible or practical in experimental pipelines. After training, our streamlined end-to-end convolutional neural network-based classification method does not require any domain specific knowledge or human intervention for quick identification of the

neuron's cell type with high accuracy. The method presented in this paper provides an efficient and standardized tool for the neuroscience community to use, thus enabling data analyses in a broader scientific context. We also showed that the network architecture learns representations that successfully distinguishes the neuron types, even when these features are not immediately recognizable upon inspection as shown by the histograms of the time series data in the Appendix.

Although we achieved the state-of-the-art results, there exists a potential source of noise in our data. The genetic labels we use in the training, validation, and test sets are currently based on the animal type used to obtain the electrophysiology recording traces. The transgenic animal lines however are not guaranteed to be accurate. It is known that some recordings obtained from a specified given type will in fact belong to another genetic type (Hu et al., 2013). Quantifying how much error is propagated from this potential noise source across all genetic types is of importance. Obtaining a more accurate genetic labels from genetic sequencing data needs to be considered in the future.

Furthermore, we plan to augment our dataset with in vivo electrophysiology recordings, which are inherently coupled with more background noise. Such an extension will be particularly beneficial for rare, but highly valuable in vivo current-clamp recordings, which are often blind to cell type. By removing the need for labor-intensive handcrafted feature computation, which traditionally requires domain expertise, our approach can enhance reproducibility and lead to faster, less biased scientific outcomes. Additionally, analyzing the network's learned features may provide insights into which neuronal activity patterns best define specific neuron types. It is widely assumed that genes encode the ion channel repertoire that underlies electrophysiological activity (Nandi et al., 2022). Identifying correlations or causal relationships between genetic and electrophysiological classifications remains a key objective in neurophysiology. This pursuit aligns with Ramón y Cajal's intuition that neuron groups possess distinct, intrinsic properties.

As foundational models continue to shape the future of machine learning and neuroscience, our approach serves not only as a current solution, but also as a potential blueprint. It provides an encoding mechanism for electrophysiology data that can seamlessly integrate into larger foundational models, ultimately bridging the gap between cellular-level neural activity and large-scale neural representations.

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

## A HISTOGRAMS OF ELECTROPHYSIOLOGY RECORDING DATA

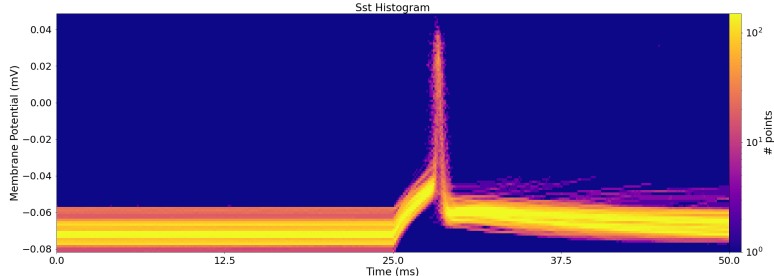

Figure 5: Time series histogram of the current-clamp electrophysiology recordings for Sst

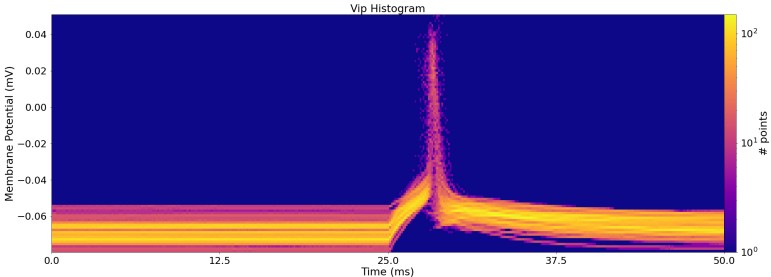

Figure 6: Time series histogram of the current-clamp electrophysiology recordings for Vip

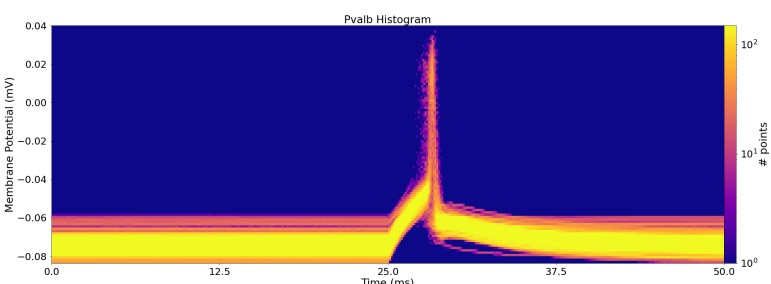

Figure 7: Time series histogram of the current-clamp electrophysiology recordings for Pvalb

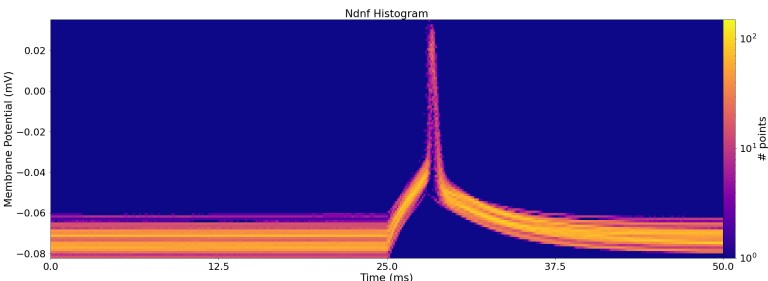

Figure 8: Time series histogram of the current-clamp electrophysiology recordings for Ndnf

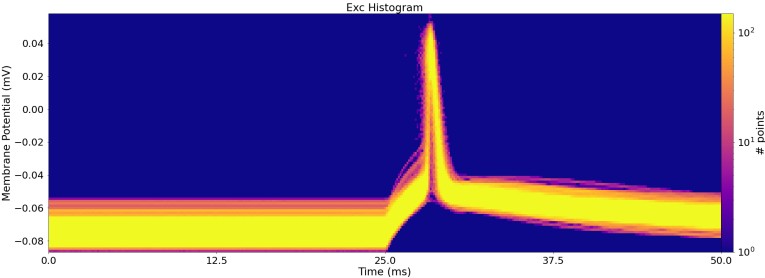

Figure 9: Time series histogram of the current-clamp electrophysiology recordings for Exc

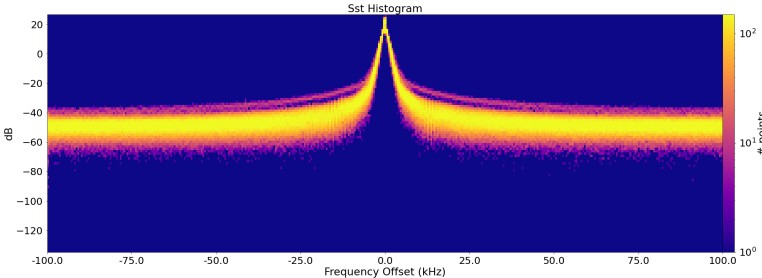

Figure 10: Fourier spectra histogram of the time series data for Sst

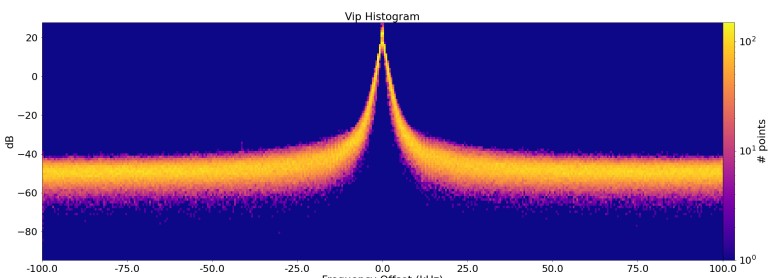

Figure 11: Fourier spectra histogram of the time series data for Vip

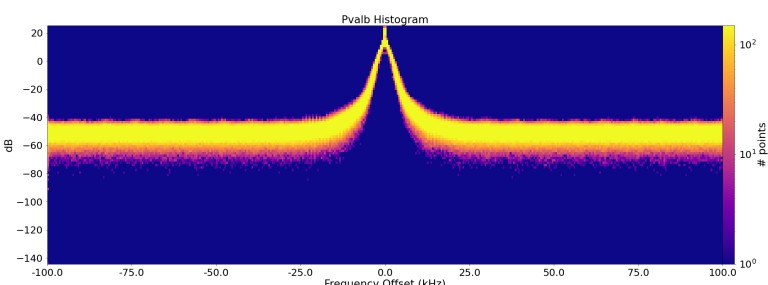

Figure 12: Fourier spectra histogram of the time series data for Pvalb

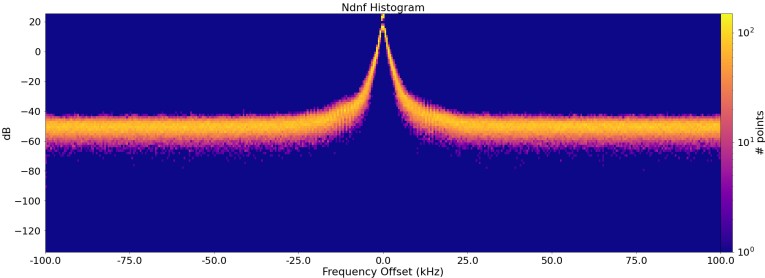

Figure 13: Fourier spectra histogram of the time series data for Ndnf

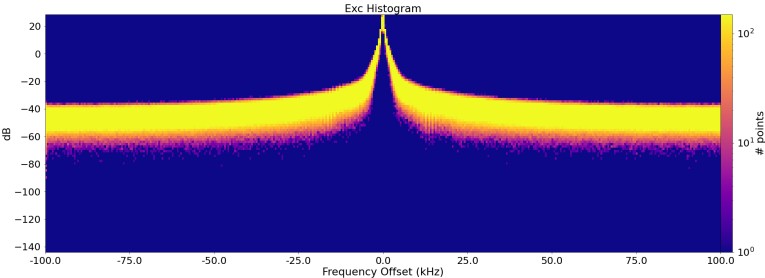

Figure 14: Fourier spectra histogram of the time series data for Exc

