# OpenReview forum: "Neocortical cell type classification from electrophysiology recordings using deep neural networks"
_ICLR.cc/2025/Conference — ICLR 2025 Conference Withdrawn Submission_

### Official Review · Reviewer_dBhH · 2024-10-31

**Soundness:** 2
**Presentation:** 2
**Contribution:** 2
**Rating:** 3
**Confidence:** 4

**Summary:**

This paper proposes a CNN-based deep learning method for classifying neocortical cell types from in-vitro patch-clamp recordings. The model takes the waveform and the real and imaginary Fourier components of a single spike (within a short window) as input. It is evaluated on the Allen Cell Types Database using both binary and 5-way classification tasks. The authors compare their method primarily with (Ghaderi et al., 2018) and claim it as the state-of-the-art in performance.

**Strengths:**

1. The paper introduces a novel model architecture that uses both raw and Fourier-processed spike data for cell type classification.
2. The model trained on a binary classification task demonstrated effective data efficiency, requiring fewer data when transferred to the more challenging 5-way classification task.

**Weaknesses:**

Major Weaknesses

1. The comparison with previous methods is limited, as the authors only benchmark against (Ghaderi et al., 2018). Other deep-learning-based methods for the same classification task, such as (Ophir et al., arXiv, 2023), are not included, which limits the ability to fully assess whether the proposed approach truly represents the state-of-the-art.
2. The performance comparison only uses overall accuracy. Given the imbalanced nature of the dataset, it would be more informative to report additional metrics, such as precision, recall, and F1 score, which are also commonly used in similar studies.
3. A potential strength of this work is that it avoids hand-crafted features, yet the authors do not empirically demonstrate whether their method significantly outperforms traditional approaches that rely on hand-crafted features.
4. The authors focus on the first spike in response to a short square pulse as input for classification, but it’s unclear why. Different neurons often exhibit distinct firing patterns with longer stimuli, including adaptation or irregular firing properties, which could provide additional classification-relevant information. A discussion on the scalability of the model to incorporate diverse response-stimuli pairs and the robustness of the model if subsequent spikes were used as input would improve clarity.
5. The authors mention that, given the same stimuli, the same neuron might exhibit variable responses, known as single-trial variability. Although the Allen Cell Types Database includes multiple trials per neuron for the same stimulus, it is unclear how the model accounts for or performs under such variability.

Minor Weaknesses

1. The manuscript could improve in clarity. For instance, presenting the ablation analysis in a table or figure format rather than in text would significantly improve readability.
2. The authors briefly mention the dataset’s imbalance problem. This is indeed a critical issue, especially as there are many subclasses within each cell type. Discussing how the model may handle or be impacted by this imbalance would provide useful insights for future work.

**Questions:**

Please refer to the weaknesses section.

---

### Official Review · Reviewer_3XJG · 2024-11-02

**Soundness:** 2
**Presentation:** 3
**Contribution:** 2
**Rating:** 3
**Confidence:** 3

**Summary:**

This paper discusses an CNN based method to classify neuron type. Through empirical results, authors demonstrate that using 1D-CNN would improve the classification performance on the dataset.

**Strengths:**

Paper is easy to follow, writing is good.

**Weaknesses:**

This paper shows that using 1D-CNN  would help improve the dataset's performance to classify neuron types. However, I am not convinced this contribution is significant enough to be presented at this conference. The only contribution is improved performance by using a 1D-CNN, which is not major.

**Questions:**

No question, as the work is straight forward.

---

### Official Review · Reviewer_xJxj · 2024-11-03

**Soundness:** 1
**Presentation:** 2
**Contribution:** 1
**Rating:** 3
**Confidence:** 4

**Summary:**

This paper introduces a cell type classification model applied to the Allen cell types datasets. It presents a CNN model that uses multiple inputs, including the raw membrane potential and Fourier components, to classify different cell types. The CNN model was shown to be effective in cell type classification, outperforming an existing baseline model.

**Strengths:**

1. This paper attempts to tackle an important problem of cell type classification in neuroscience.
2. The writing is clear and easy to follow.

**Weaknesses:**

1. This method requires more comprehensive evaluation on additional cell type classification datasets and against more baseline methods before we can conclude if it is state-of-the-art or useful.
2. The experimental setup is somewhat difficult to follow, and I am confused about how the training, validation, and test sets are partitioned. It may be important for the method to adopt more standard experimental practices commonly found in ML or computational neuroscience literature.
3. More qualitative examples are needed to help readers understand and assess the model's usefulness.

**Questions:**

Overall, I believe this method requires a more thorough evaluation and comparison with additional baselines before it can be accepted at ICLR. At this stage, the method is not fully mature and has potential for improvement in the future. I hope the following suggestions could help the author improve this paper.

**Major:**
1. The author has only used the Allen cell type datasets. **Can the method be applied to other datasets such as [1]** to make sure it is not overfitting to a single dataset?
2. The author considered only one baseline and did not conduct actual experiments for direct comparison, as noted by the author's statement in the paper, “Although not necessarily a direct comparison of performance, Ghaderi et al. (2018) achieved an overall accuracy …” **Why didn’t the author perform a direct comparison with the method in Ghaderi et al. (2018)?** Additionally, **could the author include other baselines, such as PhysMAP [2] and a VAE-based method [3]?**
3. In Section 3.2, the author mentioned "using the maximum validation accuracy over 100 epochs" and stated that "an 8:2 ratio training-validation data set split was used to select the optimal network model configuration." **I found the experimental setup and train-validation-test partitioning unclear. Could the author please clarify this?** The evaluation would be more valid if the author follow common procedures in ML or those adopted by previous work [2-3].

**Minor:**
1. In the introduction, the author mentioned that “previous approaches suffered in classification accuracy as they relied on AP shape, spiking pattern, or cell shape parameters that span a continuous feature space, which often do not have clear separation boundaries.” I find this statement confusing. Could the author elaborate on this point and explain why the proposed method is an improvement over previous approaches?
2. This paper uses raw time series data (membrane potential) and Fourier components. I wonder why single neurons waveforms were not considered. Additionally, since the CNN model takes three types of inputs, could the author provide an ablation study on the impact of each input type on model performance?
3. In Section 3.4, why does the author use the proposed transfer learning strategy to address imbalanced classification? Why not consider a simpler approach, such as upweighting the minority class during training, which is often effective? I’m curious if there’s relevant literature that I might be overlooking.

[1] Ye, Z., Shelton, A. M., Shaker, J. R., Boussard, J., Colonell, J., Birman, D., ... & Steinmetz, N. A. (2023). Ultra-high density electrodes improve detection, yield, and cell type identification in neuronal recordings. bioRxiv.

[2] Lee, E. K., Gul, A., Heller, G., Lakunina, A., Jaramillo, S., Przytycki, P., & Chandrasekaran, C. (2024). PhysMAP-interpretable in vivo neuronal cell type identification using multi-modal analysis of electrophysiological data. BioRxiv, 2024-02.

[3] Beau, M., Herzfeld, D. J., Naveros, F., Hemelt, M. E., D’Agostino, F., Oostland, M., ... & Medina, J. F. (2024). A deep-learning strategy to identify cell types across species from high-density extracellular recordings. bioRxiv.

---

### Official Review · Reviewer_UBSu · 2024-11-04

**Soundness:** 3
**Presentation:** 2
**Contribution:** 3
**Rating:** 5
**Confidence:** 5

**Summary:**

The paper addresses the challenge of classifying neuron cell types from electrophysiology recordings, specifically noting the scarcity of models and literature focused on using machine learning techniques for this classification beyond basic dimensionality reduction and clustering.

The authors introduce a convolutional neural network that achieves a low error rate on this task with minimal preprocessing. They also demonstrate the advantages of fine-tuning in handling class imbalance effectively.

**Strengths:**

This is a crucial problem with the potential to drive significant advancements in neuroscience as neuronal technologies continue to evolve. Given the recent annotation of the Allen Institute Cell Types Database, this issue is particularly worth addressing. The paper effectively explains the problem and highlights its importance.

From the results, it is evident that the architecture presented in the paper works well for cell types with sufficient enough data (i.e. E and PV cells > 90% precision) without fine-tuning and minimal pre-processing, within a 50ms window of an action potential.

**Weaknesses:**

**Accuracies are not well-presented:**

- The accuracy results should be organized into a table to improve clarity. Currently, the reader has to manually recall multiple accuracy values from related work, which disrupts readability.

**N-fold cross-validation:**
- The work could greatly benefit from N-fold cross validation, especially considering that the multi-modal architecture is relatively trivial. It would greatly strengthen results and potentially avoid choosing bad hyper-parameters.

**(Main criticism) Weak baselines:**

- Ghaderi et al, 2018 is not a sufficient baseline for evaluating the performance of the proposed model. This work in particular as stipulated in the paper, has not been evaluated on the same Allen Dataset. This makes it hard to gain perspective on the quality of the results.

- We are in the age of neuro-foundational models, it would have been a good exercise to evaluate the proposed model against transfer benchmarks on the LaBram model (from last year's ICLR spotlight - Jiang et al, 2024). This, for example, is a model that has been trained on electrophysiological recordings, in particular, EEG data. Fine-tuning the model on the Allen Cell Types dataset would be a trivial benchmark.

- I would guess that a neurofoundational model like LaBram could do better with imbalanced datasets like that of the Allen dataset, due to the pre-training of the model on EEG data despite it not being single-cell recordings. I'd be intrigued to see how this compares against your model.

**Some commentory**:

This is truly interesting work, and I believe that this has the potential to be quite useful in the field of neuroscience and NeuroAI. I think more work needs to be done in answering questions like, "why not a transformer? Is it an overkill", "how much data is enough data to solve this task?".  Your results suggest that a CNN does not need that much data (~500) to identify PV interneurons, which is an interesting find and aligns with the hypothesis that neuron groups possess distinct properties. I think this work (+ Allen Cell Dataset) makes some headway in understanding how to identify cell types but I think more time can be invested in thinking of appropriate baselines.

**Questions:**

It is unclear to me what are the implications of finetuning/pre-training on 5-class / 2-class. It seems as if pre-training on 2-class and fine-tuning on 5-class underperforms compared to training the network end-to-end on 5-classes directly, but the opposite effect is the case when pre-training on 5-class and fine-tuning on 2 classes.

* 5-class: Exc, Pvalb, Sst, Ndnf, and Vip
* 2-class: Exc, Inh

Could you clarify this result and especially it's implications in light of your contributions to the literature?

---

### Note · Authors · 2024-12-04

**Comment:**

We would like to first express our sincere gratitude to the reviewers and the organizers for considering our submission to ICLR 2025. After careful consideration, we have decided to withdraw our submission in order to take the time necessary to incorporate every valuable feedback as well as helpful suggestion for improvement provided to us by the reviewers. We plan to address all concerns pointed out by the comments regarding both the strengths and weaknesses of our approach for future work so that our submission can be made more competitive. Thank you very much once again for the helpful reviews. We will significantly improve both the impact and contribution of our work, and we will submit our work again after more work in the near future.

**Withdrawal Confirmation:**

I have read and agree with the venue's withdrawal policy on behalf of myself and my co-authors.